# Influence of a Zn Interlayer on the Interfacial Microstructures and Mechanical Properties of Arc-Sprayed Al/AZ91D Bimetals Manufactured by the Solid–Liquid Compound Casting Process

**DOI:** 10.3390/ma12193273

**Published:** 2019-10-08

**Authors:** Ke He, Jianhua Zhao, Jun Cheng, Jingjing Shangguan, Fulin Wen, Yajun Wang

**Affiliations:** 1College of Materials Science and Engineering, Chongqing University, Chongqing 400044, China; 2National Engineering Research Center for Magnesium Alloys, Chongqing University, Chongqing 400044, China

**Keywords:** solid–liquid compound casting, arc spraying, interface, microstructure, intermetallic compound, mechanical property

## Abstract

A novel technique combining solid–liquid compound casting (SLCC) with arc spraying was designed to manufacture the arc-sprayed Al/AZ91D bimetals with a Zn interlayer. The Al/Mg bimetal was produced by pouring the AZ91D melt into the molds sprayed with Al/Zn double-deck coating, during which the arc-sprayed Zn coating acted as the interlayer. The effect of the Zn interlayer on microstructures, properties, and fracture behaviors of arc-sprayed Al/AZ91D bimetals by SLCC was investigated and discussed in this study. The optimal process parameter was acquired by analyzing the results from different combinations between the arc-spraying time of the Zn coating (10, 18, and 30 s) and the preheat time of the Al/Zn double-deck coating (6 and 12 h). The interfacial microstructures of the arc-sprayed Al/AZ91D bimetals with a Zn interlayer could be approximately divided into two categories: One was mainly composed of (*α*-Mg + Al_5_Mg_11_Zn_4_) and (*α*-Al + Mg_32_(Al, Zn)_49_) structures, and the other primarily consisted of (*α*-Mg + Al_5_Mg_11_Zn_4_), (MgZn_2_ (main) + β-Zn), and (β-Zn (main) + MgZn_2_) structures. In the interface zone, the (*α*-Mg + Al_5_Mg_11_Zn_4_) structure was the most abundant structure, and the MgZn_2_ intermetallic compound had the highest microhardness of 327 HV. When the arc-spraying time of the Zn coating was 30 s and the preheat time of the Al/Zn double-deck coating was 6 h, the shear strength of the arc-sprayed Al/AZ91D bimetal reached 31.73 MPa. Most rupture of the arc-sprayed Al/AZ91D bimetals with a Zn interlayer occurred at the (*α*-Mg + Al_5_Mg_11_Zn_4_) structure and presented some typical features of brittle fracture.

## 1. Introduction

At present, magnesium alloy with its low density and high specific strength is considered a promising material for light-weight design, which can be vastly used in automobile, aerospace, and electronics industries [1]. Nevertheless, the poor corrosion resistance of magnesium alloy is in urgent need to be improved to expand its applications, and a variety of methods and techniques focusing on this issue are continuously emerged and developed. Traditional thermal spraying is a frequently-used method for the surface protection of magnesium alloy, but the joint between the coating and magnesium matrix, which is mainly supported by mechanical bonding and physical bonding, lacks enough adhesive strength for use in tough environments. In the study of Pardo et al. [2], the Al coatings were successfully thermal-sprayed on the surfaces of AZ31, AZ80, and AZ91D magnesium alloys, but there was barely an occurrence of diffusion and reaction between the coating and matrix metals. A similar phenomenon could be found within the interface between the substrates and coatings in a study about cold spraying, arc spraying, and plasma spraying [3].

Due to the formation of new phases which should not exist in the matrix metals, chemical bonding and metallurgical bonding within the interface could be achieved by the compound casting process. Li et al. [4] found that the intermetallic compounds or solid solutions of CuAl_2_, CuAl, Cu_9_Al_4_, Cu_3_Al_2_, and Cu_3_Al formed in the transitional zone of Cu/Al composite by solid–liquid compound casting (SLCC). Jiang et al. [5] discovered metallurgical bonding in steel/Al bimetallic materials by a compound casting process, and Fe_2_Al_5_, FeAl_3_, Al_8_Fe_2_Si, and Al_2_Fe_3_Si_3_ compounds were detected within the reaction layer. Ho et al. [6] also applied compound casting to acquire Cu/steel bimetal, and the interface layer consisted of carbon and CuFeO_2_ compound. Zhao et al. [7,8] were the first to combine the SLCC process with the arc-spraying technique, producing the arc-sprayed Al/Al, arc-sprayed Al/A356, and Al–Zn/AZ91D bimetallic materials. In the previous study [9], this method was also adopted in the manufacture of arc-sprayed Al/AZ91D bimetals, which utilized the arc-sprayed Al coating to prevent the outer surfaces of the AZ91D matrix from being oxidized and corroded.

For now, the dissimilar joining of magnesium alloy to aluminum alloy can be accomplished by welding and compound casting. Rao et al. [10] reported that a continuous layer of Al–Mg intermetallic compounds reduced the welding strength in friction stir spot welding of cast Mg alloy to wrought Al alloy. Liu et al. [11] revealed that the Al_3_Mg_2_ and Al_12_Mg_17_ compounds in laser welded bonding of magnesium to aluminum were of high hardness and brittleness, directly resulting in the formation of weld cracks. Hajjari et al. [12] observed that the fracture occurred within the (Al_12_Mg_17_ + Mg) eutectic structure in Al/Mg couples by the compound casting process. In general, the adhesive strength between the magnesium and aluminum metals was principally limited and hindered by the existence of Al–Mg intermetallic compounds within the interface. 

The design of a metallic interlayer added into the SLCC process, which was seldom involved and discussed before, was a novel attempt to improve the bonding between the magnesium and aluminum metals, which drew inspirations from the diffusion bonding of Al–Mg with an Ni interlayer [13], and the arc-assisted ultrasonic seam welding of Al–Mg with an Sn interlayer [14]. The addition of interlayers in various Al–Mg dissimilar welding was aimed at mitigating the Al–Mg intermetallic compounds [15]. The base and interlayer metals could be easily melted and mixed together under the effect of high temperature during the welding process. However, the Mg casting temperature is much lower than the Al–Mg welding temperature, so it is difficult to realize the joint of solid metal to interlayer and the joint of liquid metal to interlayer at the same time by SLCC. The placement of the metallic interlayer, determining the contact between the solid metal and the interlayer, also seems to be a formidable challenge in the process. Zinc has a slightly lower melting temperature than both magnesium and aluminum, which might be beneficial for the fusion and mixture of these dissimilar metals. Zinc not only has a similar crystal structure with magnesium, but also possesses a considerable solid solubility in aluminum. Moreover, there are many kinds of metallurgical reactions and chemical reactions among Al, Mg, and Zn elements under some specific conditions. As a result, zinc was eventually selected as the interlayer metal of the arc-sprayed Al/AZ91D bimetals by the SLCC process. 

In this study, it was a new exploration that the arc-spraying technique was used to prepare both a solid matrix and a metallic interlayer in the whole process. The arc-spraying time of the Zn coating and the preheat time of the Al/Zn double-deck coating were two main variables, which were in connection with the formation of microstructure and the evolution of the interface zone. Hence, the major target of this current work was to investigate the effects of the Zn interlayer on the interfacial microstructures and mechanical properties in arc-sprayed Al/AZ91D bimetals by the SLCC process.

## 2. Experimental Procedures

The raw materials for arc-spraying employed in this study were pure aluminum wires with a diameter of 2 mm and pure zinc wires with a diameter of 2 mm. The AZ91D, which is one of most widely used magnesium alloys, was chosen as the matrix metal in the casting process, and the weight of each ingot was 920 ± 60 g. The chemical compositions of aluminum wires, zinc wires, and AZ91D ingots are displayed in Table 1. In order to spray coatings on the cavity surfaces, the casting molds were designed as an assembled structure which could be separated into two symmetrical parts, and they were made of H13 steel. 

The casting molds were first brushed to remove the dirt and impurities, and placed into a box-type furnace at 523 K for 2 h. Afterwards, the preheated casting molds were immobilized on a worktable, and spread with some mold release agent. A layer of Al coating was deposited uniformly on the cavity surfaces of the molds by arc-spraying, and then a layer of Zn coating was deposited uniformly on the surfaces of the Al coating by arc-spraying, forming an Al/Zn double-deck coating on the molds. The technical parameters of the arc-spraying are listed in Table 2, and the arc-spraying time of the Zn coating on each half mold had three values (10, 18, and 30 s), making arc-sprayed Zn coatings with different average thicknesses (250, 440, and 720 μm). The arc-sprayed Al coating acted as the solid metal, while the arc-sprayed Zn coating served as the metallic interlayer. After that, the molds deposited with Al/Zn double-deck coating were put back into the furnace at 523 K for two time periods (6 and 12 h). The details about arc-spraying time of the Zn coating on each half mold and the preheat time of the Al/Zn double-deck coating are shown in Table 3.

The AZ91D ingots were placed in steel crucibles, and they were heated and melted in an electrical resistance furnace. During the fusion process, the AZ91D melt was covered by some RJ-2 flux to prevent the magnesium alloy from being oxidized and combusted. The molds deposited with Al/Zn double-deck coating were taken out from the furnace after a specific period of preheat treatment, and next the AZ91D melt was poured into the molds at 993 K. A few hours later, the magnesium alloys were completely solidified and cooled down, and the arc-sprayed Al/AZ91D bimetals with a Zn interlayer were finally acquired after the molds were removed. The whole process of the outer cladding by SLCC with a metallic interlayer is illustrated in Figure 1.

Metallographic samples were taken from the arc-sprayed Al/AZ91D bimetals by using an electrical discharge machine (EDM) to investigate the interfacial microstructure. The metallographic samples were grounded and polished with silicon carbide paper and diamond pastes, and they were etched by a solution of 3% concentrated nitric acid (HNO_3_) and 97% (vol.%) ethyl alcohol. The interfacial microstructures and fracture surfaces of the arc-sprayed Al/AZ91D bimetals with a Zn interlayer were observed by using a TESCAN VEGA III LMH scanning electron microscope (SEM, TESCAN, BrNo, South Moravia, Czech Republic) equipped with an energy dispersive X-ray spectroscope (EDS, Oxford instruments, Oxford, UK). The phase constitutions of the interface zone were detected by a D/max 2500PC X-ray diffraction (XRD, Rigaku, Tokyo, Japan). The Vickers microhardness across the interface of the arc-sprayed Al/AZ91D bimetals with a Zn interlayer was measured by a Struers Duramin-5 Microhardness Tester (Struers Inc., Cleveland, OH, USA) with a load of 100 g and a holding time of 10 s. From the AZ91D matrix to the Al coating, the indentations were arranged in a line across the whole interface zone, and suitable distances between the adjacent indentations were reserved to reduce the interference. The push out tests were conducted to determine the shear strength of the arc-sprayed Al/AZ91D bimetals with a Zn interlayer, which could represent the bonding strength between AZ91D and the Al coating. The samples for the push out tests were machined as rectangular solids, and the dimensions of them were 12 × 12 × 35 mm. The central regions of the samples were pushed by a steel punch at a displacement rate of 0.5 mm/min, and the main microstructure of the loading regions was the as-cast AZ91D alloy. The relationship curves between the shear stress and deformation displacement could be obtained, and the shear strengths of the arc-sprayed Al/AZ91D bimetals with a Zn interlayer could be also calculated using the following equation:(1)$$\delta_{\tau }=\frac{F_{\max}}{S}$$ where δ_τ_ is the shear strength, F_max_ is the maximum load, and S is the total area of the fracture surfaces. The schematic illustration of the push out test is exhibited in Figure 2, and at least three samples of each group were tested to guarantee the repeatability of the results.

## 3. Experimental Results and Discussion

### 3.1. Compositions and Microstructures

The SEM micrograph of the interfacial microstructure in the arc-sprayed Al/AZ91D bimetals with a Zn interlayer, under the condition that the arc-spraying time of the Zn coating was 10 s and the preheat time of the Al/Zn double-deck coating was 6 h (S10H6), and the corresponding EDS line scan spectrum are shown in Figure 3a,b, respectively. It can be seen that the aluminum, magnesium, and zinc elements spread into the intermediate area between the coating and the matrix during the SLCC process, forming a small and non-uniform interface zone. There were also a large number of cracks and pores existing in the interface zone between the AZ91D matrix and the Al coating. If those cracks and pores were generated during the contact between the AZ91D melt and the Al/Zn coating, these defects would become a barrier for the subsequent diffusion and reaction. As a consequence, it can be deduced that the cracks and pores at the interface formed after the element diffusion and chemical reactions.

When the arc-spraying time of the Zn coating was 10 s and the preheat time of the Al/Zn coating was 12 h (S10H12), the thickness of the interface zone in the arc-sprayed Al/AZ91D bimetals with a Zn interlayer was about 265 μm. As the preheat time of the Al/Zn coating was increased to 12 h, the Al element had more time to diffuse into the Zn coating, forming the Al or Zn solid solution. The SEM micrographs of the S10H12 sample and the corresponding EDS line scan spectrum (Figure 4) reveal that there was a homogeneous layer with a high content of aluminum at the interface zone, accompanied by some cracks and pores. The EDS point scan spectra of points “1” and “2” (marked in Figure 4b) are displayed in Figure 4d,e, and the interface zone between the AZ91D matrix and the Al coating was mainly composed of Al solid solution and Mg_32_(Al, Zn)_49_ intermetallic compound. In addition, the contents of aluminum and zinc were slightly more than that of magnesium within the interface. According to the Al–Mg–Zn ternary system, the α-Al and Mg_32_(Al, Zn)_49_ were formed through the following reaction [16]: (2)$$L\overset{762K}{\to }\;\alpha -\mathrm{Al}\;+\;{\mathrm{Mg}}_{32}{\left(\mathrm{Al},\;\mathrm{Zn}\right)}_{49}$$

During the casting process, the Zn side of the arc-sprayed coating was the first to have contact with the AZ91D melt. After solidification, the area adjacent to the AZ91D matrix was occupied by α-Al solid solutions, which dissolved a certain amount of Zn element. The enrichment of Mg and Zn atoms, combined with a certain number of Al atoms, formed the Mg_32_(Al, Zn)_49_ intermetallic compound. Therefore, the arc-sprayed Zn coating was transformed into Al–Zn coating during the long period (12 h) of preheat treatment, and Mg element diffused from the AZ91D melt into the Al–Zn coating during the SLCC process. Although the Mg element distributed throughout the whole interface zone, the Al–Mg binary intermetallic compound could not be discovered in the S10H12 sample. It could be deduced then that the presence of Zn can facilitate the formation of Al–Mg–Zn ternary intermetallic compounds instead of the Al–Mg binary intermetallic compounds in arc-sprayed Al/AZ91D bimetals. 

When the arc-spraying time of the Zn coating was 18 s, and the preheat time of the Al/Zn coating was 6 h (S18H6), the thickness of the interface zone in the arc-sprayed Al/AZ91D bimetals with a Zn interlayer was about 912 μm. As can be seen in Figure 5a, the microstructure of the interface zone can be divided into three different layers (I, II, and III) without obvious cracks or pores. Through a comprehensive analysis of the EDS results (Table 4 and Figure 9a) and XRD results (Figure 10a), the major constituents of layer I adjacent to the AZ91D matrix were α-Mg solid solution and Al_5_Mg_11_Zn_4_ intermetallic compound, and layer III adjacent to the Al coating was mainly composed of α-Al solid solution and Mg_32_(Al, Zn)_49_ intermetallic compound—while layer II between layer I and III contained β-Zn solid solution, and Al_5_Mg_11_Zn_4_ and MgZn_2_ compounds. The diffusion of Al element from the Al coating to the Zn coating was insufficient due to the limited preheat time, so the Zn coating could not completely transform into Al–Zn coating. Although the Mg atoms in the AZ91D melt had a stronger diffusion ability than Al atoms in the Al coating, the Mg atoms were inadequate to combine with all Zn atoms, leaving behind some β-Zn solid solutions in layer II (as presented in Figure 5c).

In the region near the AZ91D melt, the relative content of magnesium was high, while the relative content of aluminum was low. During the solidification process, the reactions might occur in the following order [16,17,18]:(3)$$L\overset{660\;K}{\to }\;\alpha -\mathrm{Mg}\;+\;{\mathrm{Al}}_{5}{\mathrm{Mg}}_{11}{\mathrm{Zn}}_{4}$$
(4)$$L\overset{603\;K}{\to }\;\alpha -\mathrm{Mg}\;+\;\mathrm{MgZn}+\;{\mathrm{Mg}}_{32}{\left(\mathrm{Al},\;\mathrm{Zn}\right)}_{49}$$

The *α*-Mg solid solution and Al_5_Mg_11_Zn_4_ ternary compound would precipitate in the first place, but their respective amounts were determined by the distribution of Al, Mg, and Zn elements. The zinc diffused into the AZ91D melt, and a small quantity of Al_5_Mg_11_Zn_4_ compounds as the secondary phase distributed in the AZ91D matrix adjacent to the interface zone. As the relative content of magnesium decreased in layer I, α-Mg solid solutions as the secondary phase distributed in Al_5_Mg_11_Zn_4_ ternary compound. When the content of magnesium was too low to support the formation of α-Mg solid solution, a layer of Al_5_Mg_11_Zn_4_ compound was generated, as shown in Figure 5c. The content of aluminum in the AZ91D melt was low, and the diffusion of Al element from the Al coating to the Zn coating was also insufficient. Thus, there would exist a region with a low relative content of aluminum and high relative contents of magnesium and zinc. Based on the Al–Mg–Zn ternary system, an Mg–Zn intermetallic compound would come into being by the reaction below [19]: (5)$$L\overset{733\;K}{\to }\;{\mathrm{MgZn}}_{2}$$

The existence of MgZn_2_ intermetallic compound in layer II could confirm the occurrence of this reaction during the casting process. As the relative content of zinc increased continuously, another Reaction (6) would take place [19]:(6)$$L\overset{623\;K}{\to }\;{\mathrm{MgZn}}_{2}+\beta -\mathrm{Zn}$$

A number of β-Zn solid solutions precipitated and distributed in the MgZn_2_ intermetallic compound. In the region near the Al coating, the relative content of aluminum increased sharply, while the relative content of magnesium decreased rapidly, which had a similar situation with the interface zone of the S10H12 sample. The (*α*-Al + Mg_32_(Al, Zn)_49_) structure, which constituted layer III, was generated on the basis of the Reaction (2).

When the arc-spraying time of the Zn coating was 18 s and the preheat time of the Al/Zn coating was 12 h (S18H12), the thickness of the interface zone in the arc-sprayed Al/AZ91D bimetals with a Zn interlayer was about 838 μm. Figure 6 shows that the microstructure of the interface zone could be divided into two layers (I and II) without obvious cracks or pores. The EDS results (Table 5 and Figure 9b) and XRD results (Figure 10b) imply that layer I next to the AZ91D matrix comprised *α*-Mg solid solution and Al_5_Mg_11_Zn_4_ ternary compound, and the main constitutions of layer II adjacent to the Al coating were *α*-Al solid solution and Mg_32_(Al, Zn)_49_ ternary compound. What is more, the tiny *α*-Al solid solution dispersed in Mg_32_(Al, Zn)_49_ ternary compound within layer II, and there was a small amount of Al_5_Mg_11_Zn_4_ ternary compound existing between layer I and II. Different from the interfacial microstructure in the S18H6 sample, there was no MgZn_2_ intermetallic compound and β-Zn in the interface zone of the S18H12 sample. Before preheat treatment, the thickness of the Zn coating in S18H12 should be roughly equivalent to that in S18H6. Under the diffusion effect of aluminum during the preheat treatment, the Zn coating was completely transformed into the Al–Zn coating, and thus there was no area where the zinc element was enriched in the arc-sprayed coating. In the interface zone, the Mg–Zn intermetallic compound was difficult to form, while the Al–Mg–Zn ternary compound became the main component.

When the arc-spraying time of the Zn coating was 30 s and the preheat time of the Al/Zn coating was 6 h (S30H6), the thickness of interface zone in the arc-sprayed Al/AZ91D bimetals with a Zn interlayer was about 1207 μm. Figure 7 demonstrates that the microstructure of the interface zone can be divided into three layers without obvious cracks or pores. Considering the EDS results (Table 6 and Figure 9c) and XRD results (Figure 10c), it could be deduced that the main compositions of layer I adjacent to the AZ91D matrix were *α*-Mg solid solution and Al_5_Mg_11_Zn_4_ ternary compound, and β-Zn solid solution and MgZn_2_ intermetallic compound constituted layers II and III. In the region near the arc-sprayed Al coating, the reaction during the solidification process would occur based on the following equation [17,19]: (7)$$L\overset{853\;K}{\to }\;\alpha -\mathrm{Al}\;$$

The relative content of aluminum was at a high level, and zinc can dissolve in α-Al phase at a high solubility. With the increase of the distance from the Al coating, the relative content of aluminum gradually reduced, while the relative content of zinc augmented. The *α*-Al solid solution and MgZn_2_ intermetallic compound would form in accordance with the following reaction [17,19]:(8)$$L\overset{753\;K}{\to }\;{\alpha -\mathrm{Al}\;+\;\mathrm{MgZn}}_{2}$$

When the relative content of aluminum continued decreasing, and the relative content of zinc went up, a further Reaction (9) would take place during the solidification process [17,19]:(9)$$L\overset{610\;K}{\to }\;{\alpha -\mathrm{Al}\;+\;\mathrm{MgZn}}_{2}+\beta -\mathrm{Zn}$$

From that moment, the β-Zn solid solution began to precipitate with the MgZn_2_ intermetallic compound. When the content of zinc was lower than that of magnesium, the production of β-Zn solid solution was less than that of MgZn_2_ intermetallic compound, and β-Zn solid solution as the secondary phase dispersedly distributed in MgZn_2_ intermetallic compound is shown in Figure 7c,d. When the content of zinc was higher than that of magnesium, the production of β-Zn solid solution was more than that of MgZn_2_ intermetallic compound, as shown in Figure 7f. 

A thick Zn interlayer experienced a short period (6 h) of preheat treatment in the S30H6 group, which limited the diffusion time of aluminum from Al coating to Zn coating. Through the diffusion, the Al element in the *α*-Mg solid solution and Al_5_Mg_11_Zn_4_ compounds of layer I came from two sources: The AZ91D alloy and arc-sprayed Al coating. However, there still existed a region with low relative content of aluminum between the AZ91D matrix and the Al coating due to the insufficient diffusion. In this region, the content of magnesium and zinc was high, and it was easy to form the β-Zn solid solution and MgZn_2_ intermetallic compound, which constituted layers II and III. Besides, it can be seen in Figure 7e that there was a small amount of (*α*-Mg + Al_5_Mg_11_Zn_4_) structure forming between layer II and III, in which the ratio of Zn content to Mg content was close to 2:1.

When the arc-spraying time of the Zn coating was 30 s and the preheat time of the Al/Zn coating was 12 h (S30H12), the thickness of the interface zone in the arc-sprayed Al/AZ91D bimetals with a Zn interlayer was about 1560 μm. Figure 8 shows that the microstructure of the interface zone can be divided into three layers without obvious cracks or pores. According to the EDS results (Table 7 and Figure 9d) and XRD results (Figure 10d), layer I adjacent to the AZ91D matrix was mainly composed of *α*-Mg solid solution and Al_5_Mg_11_Zn_4_ ternary compound, and layers II and III were mainly constituted by β-Zn solid solution and MgZn_2_ intermetallic compound. Compared with the interfacial microstructure of the S30H6 sample, the thickness of layer I had a significant increase in the S30H12 sample, while the thickness of layer II had a reduction. After a long period (12 h) of preheat treatment, the aluminum element could sufficiently diffuse from the Al coating into the Zn coating. In the SLCC process, the magnesium element could diffuse from the AZ91D melt into the Zn coating as well. As a result, the Zn-enriched region where the relative content of aluminum was low greatly reduced in the interface zone. In the case that the aluminum content was adequate, the (*α*-Mg + Al_5_Mg_11_Zn_4_) structure was more likely to be generated within the interface, which would inhibit the formation of β-Zn solid solution and MgZn_2_ intermetallic compound.

On the whole, the SLCC process facilitated the formation of the interface zone, representing the existence of metallurgical bonding and chemical bonding between the as-cast AZ91D and the arc-sprayed Al coating. The Zn interlayer avoided the generation of Al–Mg binary compounds, and promoted the production of the Al–Mg–Zn ternary compounds. In the S18H6, S18H12, S30H6, and S30H12 samples, the (*α*-Mg + Al_5_Mg_11_Zn_4_) structure always occupied the largest area in the interface zones. When the relative content of aluminum was low, the MgZn_2_ intermetallic compounds appeared in the interface of the arc-sprayed Al/AZ91D bimetals with a Zn interlayer. 

### 3.2. Interfacial Microstructure Evolution

The coefficient of linear thermal expansion (CLTE) of zinc was 36 × 10^−6^ K^−1^, which was 38.5% and 56.5% larger than magnesium and aluminum, respectively. Therefore, zinc was very sensitive to the temperature variation in the heating and cooling stages during SLCC, resulting in a large volume change of itself. If the times when the preheat treatment was too short or the Zn coating was too thick, some areas of the Al/Zn coating would still contained a high content of zinc. On the one hand, the zinc melted ahead of the aluminum during the heating stage, and the volume growth rate of zinc was also larger than those of aluminum and magnesium. The zinc quickly expanded towards both sides, and the rapid expansion produced an outward pressure on the aluminum and magnesium, which accelerated the diffusion of zinc to the Al coating and the AZ91D melt, respectively. On the other hand, the zinc solidified after the aluminum and magnesium in the cooling stage, and the volume shrinkage rate of zinc was larger than those of aluminum and magnesium. Similarly, the fast contraction process of zinc also produced an inward pressure, which caused the generation of cracks and pores within the interface.

In addition, some pores formed in the coating during the arc-spraying process, and those pores acted as the channels for fusion and diffusion, improving the bonding between the AZ91D matrix and the Al coating. Nevertheless, those pores also become the sources of cracks under the effect of internal stress during cooling and solidification. If the time of the preheat treatment was too short or the Zn coating was too thick, the MgZn_2_ intermetallic compound with high brittleness and hardness could be generated in the interface zone, which could reduce the bonding strength between Al coating and AZ91D matrix. In the interface zone of the S10H12 sample, the Mg_32_(Al, Zn)_49_ ternary compound with high brittleness and hardness induced the formation of cracks during the solidification process.

Figure 11 reveals the microstructure evolution of the Al/Zn double-deck coating under different times of preheat treatment, bringing about the transformation from Al/Zn coating to *x*Al-(1 − *x*) zinc coating. When *x* was greater than 50%, the main composition of the arc-sprayed coating was aluminum solid solution. When *x* was less than 50%, the major component of the arc-sprayed coating was zinc solid solution. The value of x decreased with the increase of the distance from the Al coating, which conformed to the discipline of diffusion gradient. After different times of preheat treatment, the microstructures of the Al/Zn double-deck coating could be divided into two cases: (1) The whole Zn coating transformed to *x*Al-(1 − *x*)Zn coating (*x* > 50%), (2) partial Zn coating near the Al coating transformed to *x*Al-(1 − *x*)Zn coating (*x* > 50%) and the remaining part transformed to *x*Al-(1 − *x*)Zn coating (*x* < 50%). Therefore, the subsequent formation process of interface zones could also be summarized into two different categories (as shown in Figure 12 and Figure 13).

When the whole Zn coating transformed to *x*Al-(1 − *x*)Zn coating (*x* > 50%), the distributions of aluminum and zinc in the Al/Zn coating were relatively uniform before casting. During the SLCC process, the Al, Mg, and Zn elements had full contact with each other, and the interface zone mainly consisted of the Al–Mg–Zn ternary compound. The relative content of magnesium was high on the side near the AZ91D matrix, and a great deal of (*α*-Mg + Al_5_Mg_11_Zn_4_) structures were generated in that area. The relative content of aluminum was high on the side near the arc-sprayed Al coating, and a mass of (*α*-Al + Mg_32_(Al, Zn)_49_) structures formed in that region. As a consequence, the interface zone was mainly composed of the (*α*-Mg + Al_5_Mg_11_Zn_4_) and (*α*-Al + Mg_32_(Al, Zn)_49_) structures.

When partial Zn coating near the Al coating transformed to *x*Al-(1 – *x*)Zn coating (*x* > 50%) and the remaining part transformed to *x*Al-(1 − *x*)Zn coating (*x* < 50%), the distributions of aluminum and zinc elements in the Al/Zn coating were relatively non-uniform before casting. The zinc mainly distributed on the side which had contact with the AZ91D melt, while aluminum primarily distributed on the side near the Al coating. During the SLCC process, the magnesium and aluminum in the AZ91D melt had contact with zinc, generating the (*α*-Mg + Al_5_Mg_11_Zn_4_) structure first. As the relative content of magnesium and aluminum decreased, the relative content of zinc increased, and then the MgZn_2_ intermetallic compound and β-Zn solid solution began to precipitate within the interface. Hence, the interface zone was principally made up of the (*α*-Mg + Al_5_Mg_11_Zn_4_), (MgZn_2_ (main) + β-Zn), and (β-Zn (main) + MgZn_2_) structures.

### 3.3. Microhardness Distribution

The microhardness distributions across the interfaces of the arc-sprayed Al/AZ91D bimetals with a Zn interlayer under different arc-spraying times of Zn coating (18 and 30 s) and various preheat times of the Al/Zn double-deck coating (6 and 12 h) are demonstrated in Figure 14. On both sides of the interface zone, the arc-sprayed Al coating and the AZ91D matrix had stable microhardness values which maintained in a relatively small range, and their average microhardness was approximately 50 HV and 64 HV, respectively. On the contrary, the interface zones of the S18H6, S18H12, S30H6, and S30H12 samples presented a relatively high level of microhardness. Layer I, which was mainly composed of (*α*-Mg + Al_5_Mg_11_Zn_4_) structure, had an average hardness of 188 HV. The MgZn_2_ intermetallic compounds in the interface zones of the S18H6, S30H6, and S30H12 samples had the highest average hardness value of 327 HV, which show as the highest peaks in Figure 14a,c,d. Liu et al. [20] also found that the microhardness value experienced an enormous increase at the layer of MgZn_2_ intermetallic compound in the gas tungsten arc butt welding joint of Mg–Al with Zn filler metal. What is more, the microhardness value of the region between the (*α*-Mg + Al_5_Mg_11_Zn_4_) structure and Al_5_Mg_11_Zn_4_ ternary compound slightly decreased compared with the microstructure on its two sides. In the S18H12 sample, there was no MgZn_2_ intermetallic compound with high hardness existing in the interface zone. The region near the Al coating, which consisted of (*α*-Al + Mg_32_(Al, Zn)_49_) structure, had an average hardness value of 111 HV. In the S30H6 and S30H12 samples, the region near the Al coating was mainly composed of a large amount of β-Zn solid solution and a small amount of MgZn_2_ intermetallic compound. Although the microhardness of MgZn_2_ intermetallic compound was very high, the average hardness value of the (β-Zn (main) + MgZn_2_) structure was only about 102 HV. In general, the hardness of the (*α*-Al + Mg_32_(Al, Zn)_49_) structure was lower than that of the (*α*-Mg + Al_5_Mg_11_Zn_4_) structure within the interface of the arc-sprayed Al/AZ91D bimetals with a Zn interlayer.

### 3.4. Shear Strength and Fractography

In this study, the shear strength was used to evaluate the bonding strength between the arc-sprayed Al coating and the AZ91D matrix. Figure 15 displays the relationship curves between the shear stress and deformation displacement arc-sprayed Al/AZ91D bimetals with a Zn interlayer under different arc-spraying times of the Zn coating (18 and 30 s) and various preheat times of the Al/Zn double-deck coating (6 and 12 h). 

By comparing the results of the S30H6 and S30H12 samples, it can be found that the preheat time of the Al/Zn double-deck coating had a significant influence on the shear strength of the arc-sprayed Al/AZ91D bimetals. As the preheat time of Al/Zn coating increased from 6 to 12 h, the area of the interface zone also had an increase, but the shear strength sharply dropped from 31.73 to 15.44 MPa. There was a close connection between AZ91D alloy and the arc-sprayed Al coating, and a variety of intermetallic compounds were generated at the interface, confirming the existence of chemical bonding and metallurgical bonding.

The arc-spraying time of the Zn coating was also an important factor which could affect the shear strength of the arc-sprayed Al/AZ91D bimetals, and it could directly determine the thickness of the Zn coating. When the arc-spraying time of zinc coating was short (10 s), the shear strengths of the S10H6 and S10H12 samples were 5.06 MPa and 7.23 MPa, respectively. The cracks and holes within the interface of the S10H6 and S10H12 samples led to a reduction of shear strength of the arc-sprayed Al/AZ91D bimetals with a Zn interlayer. When the preheat time of the Al/Zn double-deck coating was 6 h, the shear strength of the arc-sprayed Al/AZ91D bimetals went up with the increase of arc-spraying time of the Zn coating. Under the condition that the arc-spraying time of the Zn coating was 30 s and the preheat time of the Al/Zn double-deck coating was 6 h, the arc-sprayed Al/AZ91D bimetal with a Zn interlayer had the highest shear strength of 31.73 MPa among all the samples, which was 36.5% more than the arc-sprayed Al/AZ91D bimetal without an interlayer in previous work [9]. When the preheat time of the Al/Zn double-deck coating was 12 h, the shear strength of arc-sprayed Al/AZ91D bimetals first increased to 26.19 MPa, and then decreased to 15.44 MPa with the increase of arc-spraying time of the Zn coating.

Figure 16 and Figure 17 are the fracture analysis results of the Al/AZ91D bimetals with a Zn interlayer under different arc-spraying times of the Zn coating (18 and 30 s) and various preheat times of the Al/Zn double-deck coating (6 and 12 h). In general, there were lots of cleavage steps in the fracture surfaces of the S18H6, S18H12, and S30H12 samples, which implies that the major fracture mode was brittle fracture. According to the results of EDS point scan, the fracture mainly occurred in the (*α*-Mg + Al_5_Mg_11_Zn_4_) structure thanks to the high brittleness of Al_5_Mg_11_Zn_4_ compound. The S30H12 sample had the largest interface zone with the most (*α*-Mg + Al_5_Mg_11_Zn_4_) structure, indicating that it also had the largest area for brittle fracture. Compared with the S18H6 and S18H12 samples, there were more slopes and steps on the fracture surface of the S30H12 sample, and the shear strength of the S30H12 sample was lower. Hence, it could be concluded that the increase of intermetallic compounds with high brittleness in the interface zone, which increased the occurrence probability of brittle fracture, caused the reduction of shear strength between the AZ91D matrix and the Al coating.

Different from other samples, a mass of dimples appeared on the fracture surface of the S30H6 sample, as shown in Figure 17a. The EDS point scan result (Figure 17c) demonstrates that those dimples were located in the MgZn_2_ intermetallic compound. The MgZn_2_ intermetallic compound was a typical Laves phase with a high brittleness, which had a similar crystal structure with magnesium [21]. In the interface zone of the S30H6 sample, there was a thick layer constituted by a large amount of the MgZn_2_ intermetallic compound and a small amount of β-Zn solid solution, as displayed in Figure 7c,d. In addition, the β-Zn solid solution, which was a soft phase, dispersedly distributed in MgZn_2_ intermetallic compound with a higher hardness. Dai et al. [22] adopted the Zn interlayer in the arc-assisted ultrasonic seam welding of Al/Mg joints to replace the Al–Mg intermetallic compounds with β-Zn solid solution and MgZn_2_ intermetallic compounds in the interface zone, which effectively improved the bonding strength of the welding joints. The β-Zn solid solution was more flexible and less brittle than the MgZn_2_ intermetallic compounds. When the soft secondary phase (β-Zn solid solution) was dispersedly distributed in the hard matrix (MgZn_2_ intermetallic compound), there was a strong gravitational interaction between the secondary phase and dislocations, which effectively strengthened the interfacial microstructure [21]. Liu et al. [23] revealed that in the diffusion bonded joint of Al/Mg with Zn–5Al interlayer after a holding time of 3 s, the dispersive distribution of Al solid solution in MgZn_2_ intermetallic compounds improved the shear strength of the joint. Although there was also a certain amount of MgZn_2_ intermetallic compounds and β-Zn solid solution in the interface zones of the S18H6 and S30H12 samples, the shear strengths of them were not enhanced prominently. It was because MgZn_2_ intermetallic compounds and β-Zn solid solution distributed in layers, and the (*α*-Mg + Al_5_Mg_11_Zn_4_) structure with high brittleness occupied a large scope of the interface zone. What is more, the fracture of the S30H6 sample occurred not only in the (MgZn_2_ + β-Zn) structure, but also in the (*α*-Mg + Al_5_Mg_11_Zn_4_) structure. It can be observed in Figure 17b that the tearing ridge in the (*α*-Mg + Al_5_Mg_11_Zn_4_) structure extends along the same direction, which indicates that the S30H6 sample also experienced the brittle fracture during the shear test.

## 4. Conclusions

In this study, the effects of a Zn interlayer on interfacial microstructure and mechanical properties of arc-sprayed Al/AZ91D bimetals by SLCC were researched and discussed. The main conclusions can be summarized as follow:(1)The interfacial microstructures of the arc-sprayed Al/AZ91D bimetals with a Zn interlayer can be approximately divided into two categories: One was mainly composed of (α-Mg + Al_5_Mg_11_Zn_4_) and (α-Al + Mg_32_(Al, Zn)_49_) structures, and the other primarily consisted of (α-Mg + Al_5_Mg_11_Zn_4_), (MgZn_2_ (main) + β-Zn), and (β-Zn (main) + MgZn_2_) structures.(2)When the arc-spraying time of the Zn coating was 30 s and the preheat time of the Al/Zn double-deck coating was 12 h, the arc-sprayed Al/AZ91D bimetals with a Zn interlayer had the thickest interface zone. Metallurgical bonding and chemical bonding were achieved by the formation of new phases during the SLCC process.(3)The (α-Mg + Al_5_Mg_11_Zn_4_) structure, which occupied the largest area in the interface zones of the arc-sprayed Al/AZ91D bimetals with a Zn interlayer, had an average hardness of 188 HV, and the MgZn_2_ intermetallic compound had the highest average hardness of 327 HV.(4)The pores and cracks within the interface weakened the bonding between the Al coating and the AZ91D matrix. The dispersive distribution of Mg solid solution in MgZn_2_ intermetallic compounds strengthened the interfacial microstructure of the arc-sprayed Al/AZ91D bimetal.(5)When the arc-spraying time of the Zn coating was 30 s and the preheat time of the Al/Zn double-deck coating was 6 h, the arc-sprayed Al/AZ91D bimetals had the highest shear strength of 31.73 MPa. Most of the brittle fracture in the arc-sprayed Al/AZ91D bimetal initiated and occurred in the (α-Mg + Al_5_Mg_11_Zn_4_) structure.

## Figures and Tables

**Figure 1 materials-12-03273-f001:**
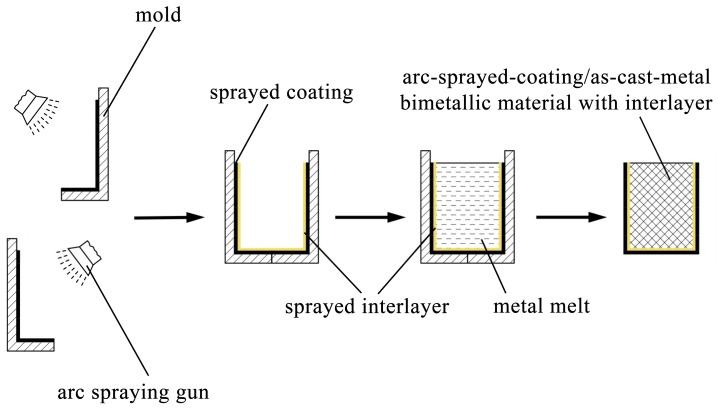
Schematic diagram of outer cladding by SLCC with a metallic interlayer.

**Figure 2 materials-12-03273-f002:**
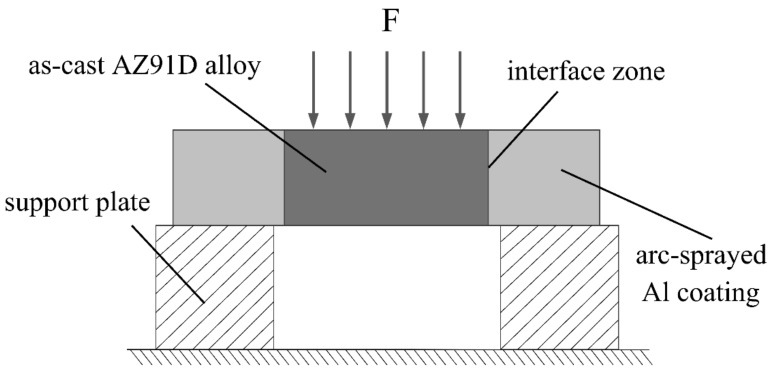
Schematic illustrations of the push out test.

**Figure 3 materials-12-03273-f003:**
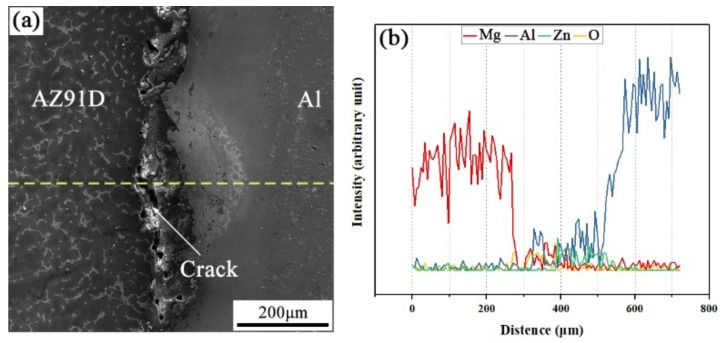
(**a**) SEM micrograph and (**b**) EDS line scan spectrum (marked as dashed line in (**a**)) of the interface in the arc-sprayed Al/AZ91D bimetals with a Zn interlayer (arc-spraying Zn for 10 s and Al/Zn double-deck coating preheated for 6 h).

**Figure 4 materials-12-03273-f004:**
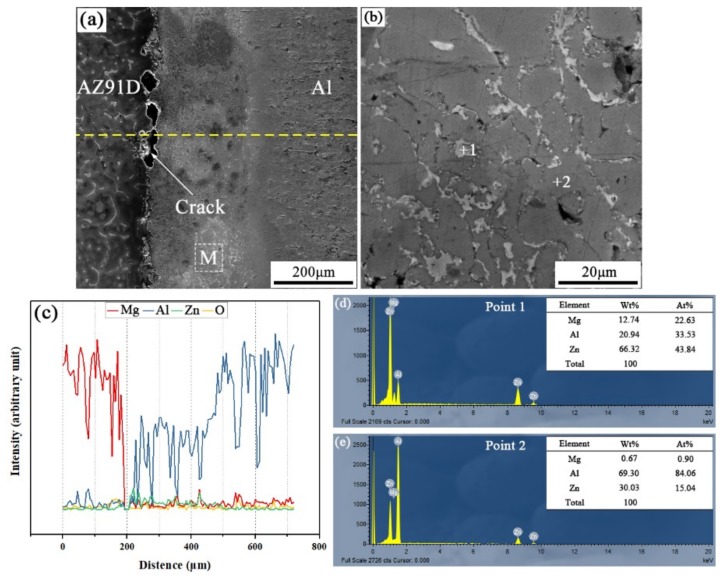
Arc-sprayed Al/AZ91D bimetals with a Zn interlayer (arc-spraying Zn for 10 s and Al/Zn double-deck coating preheated for 12 h): (**a**) SEM micrograph of interface, (**b**) SEM micrograph of area M in (**a**), (**c**) EDS line scan spectrum (marked as dashed line in (**a**)), (**d**,**e**) EDS point scan spectra of points 1 and 2 in (**b**).

**Figure 5 materials-12-03273-f005:**
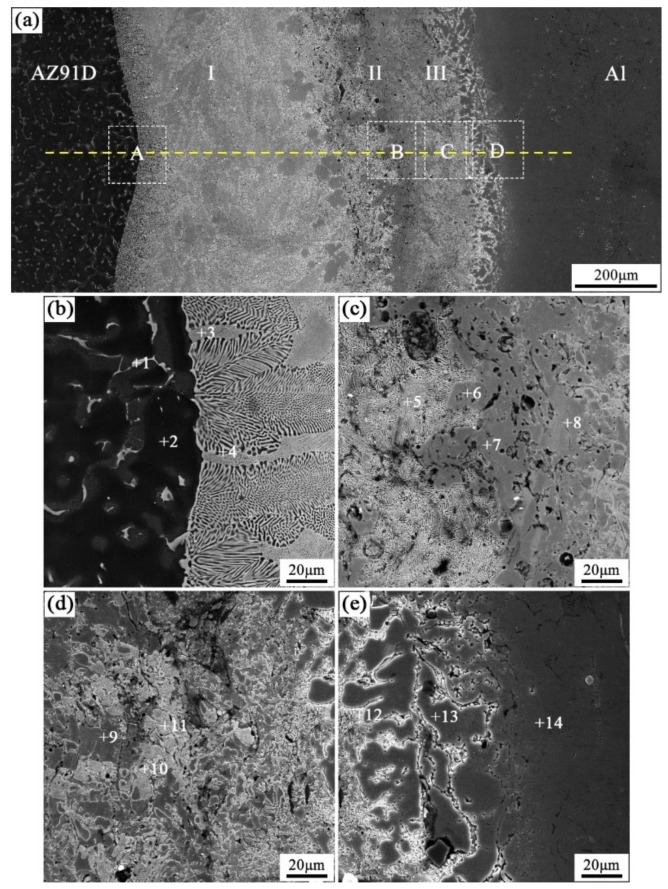
SEM micrographs of the interface in arc-sprayed Al/AZ91D with a Zn interlayer (arc-spraying Zn for 18 s and Al/Zn double-deck coating preheated for 6 h): (**a**) general view, (**b**–**e**) areas A, B, C, and D in (**a**), respectively.

**Figure 6 materials-12-03273-f006:**
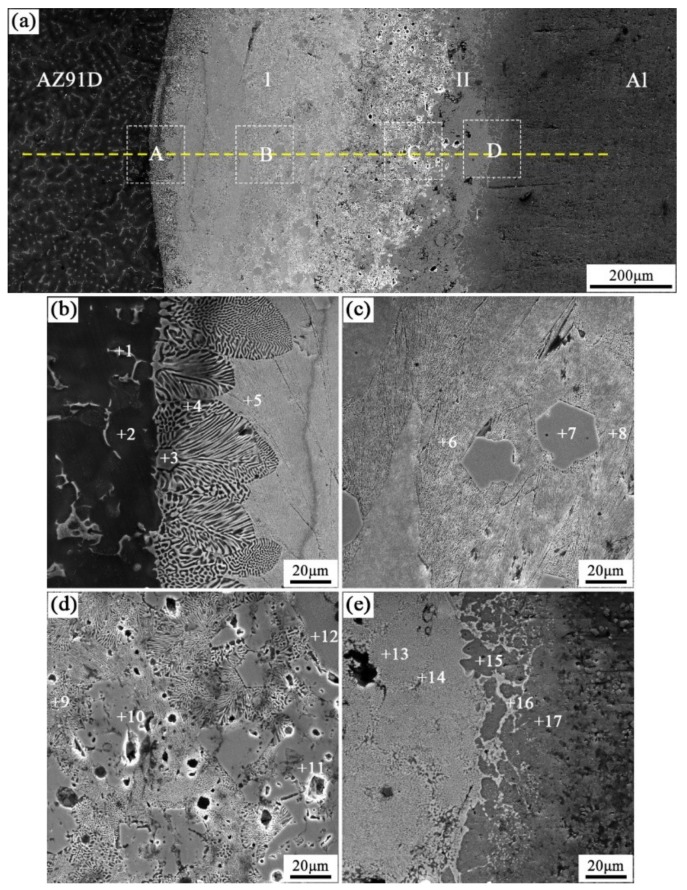
SEM micrographs of the interface in arc-sprayed Al/AZ91D with a Zn interlayer (arc-spraying Zn for 18 s and Al/Zn double-deck coating preheated for 12 h): (**a**) general view, (**b**–**e**) areas A, B, C, and D in (**a**), respectively.

**Figure 7 materials-12-03273-f007:**
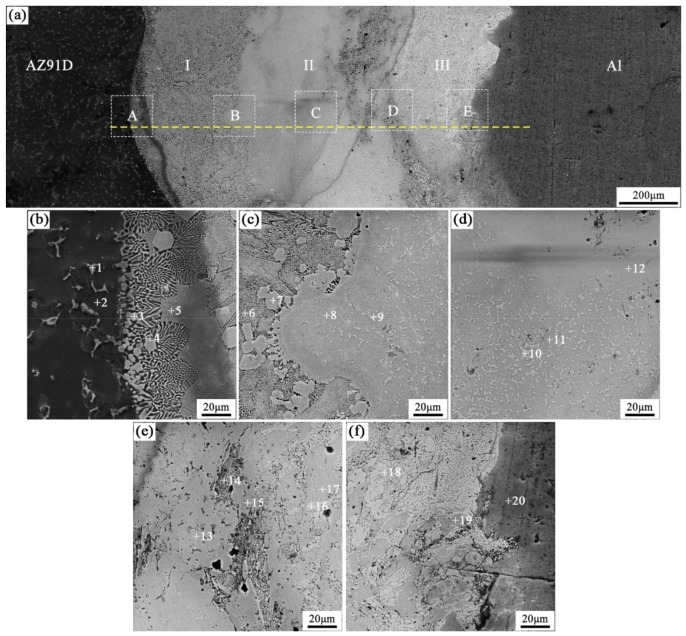
SEM micrographs of the interface in arc-sprayed Al/AZ91D with a Zn interlayer (arc-spraying Zn for 30 s and Al/Zn double-deck coating preheated for 6 h): (**a**) general view, (**b**–**f**) areas A, B, C, D, and E in (**a**), respectively.

**Figure 8 materials-12-03273-f008:**
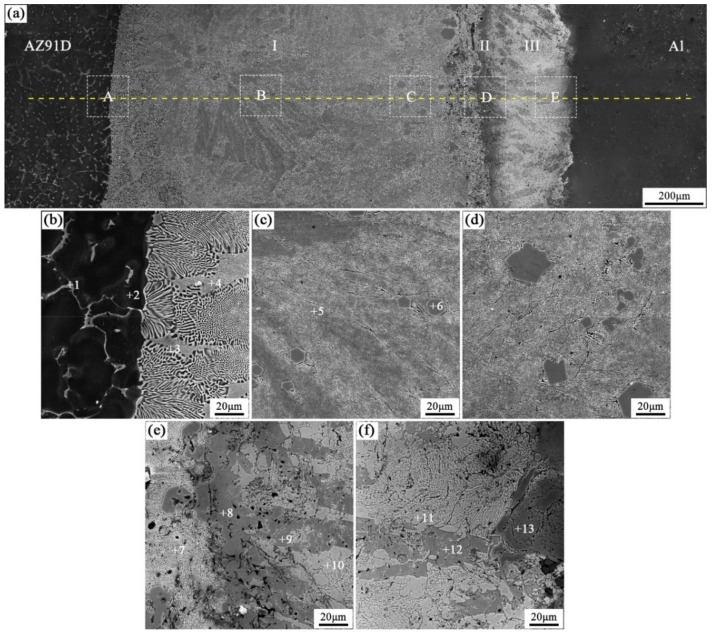
SEM micrographs of the interface in arc-sprayed Al/AZ91D with a Zn interlayer (arc-spraying Zn for 30 s and Al/Zn double-deck coating preheated for 12 h): (**a**) general view, (**b**–**f**) areas A, B, C, D, and E in (**a**), respectively.

**Figure 9 materials-12-03273-f009:**
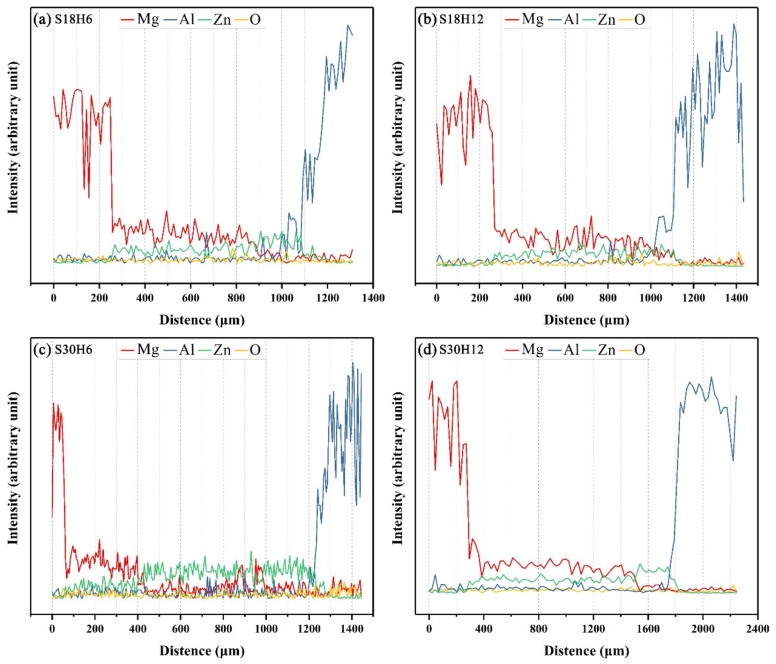
EDS line scan spectra across the interfaces of arc-sprayed Al/AZ91D with a Zn interlayer: (**a**) S18H6, (**b**) S18H12, (**c**) S30H6 and (**d**) S30H12, as marked by dashed lines in Figure 5a, Figure 6a, Figure 7a, and Figure 8a, respectively.

**Figure 10 materials-12-03273-f010:**
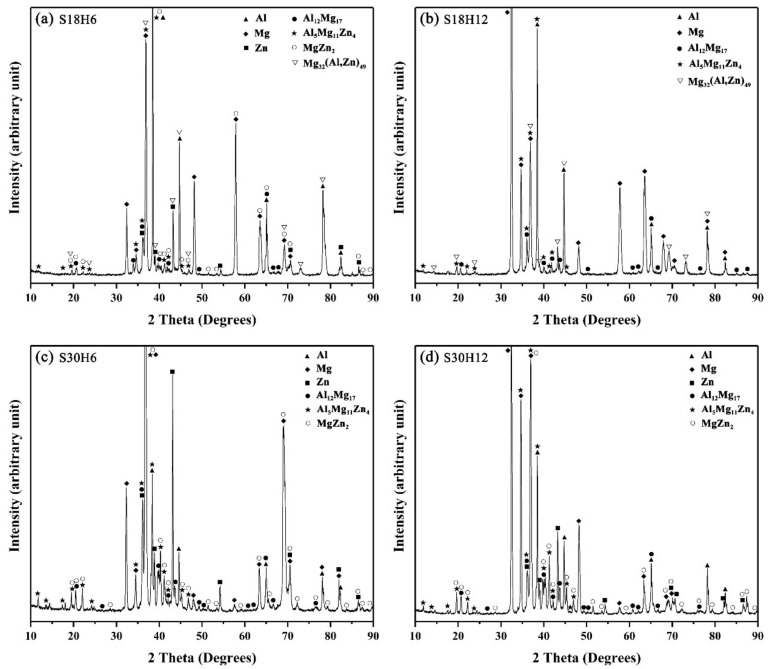
XRD pattern of the constituent phases at the interface of the arc-sprayed Al/AZ91D with a Zn interlayer: (**a**) S18H6, (**b**) S18H12, (**c**) S30H6, and (**d**) S30H12.

**Figure 11 materials-12-03273-f011:**
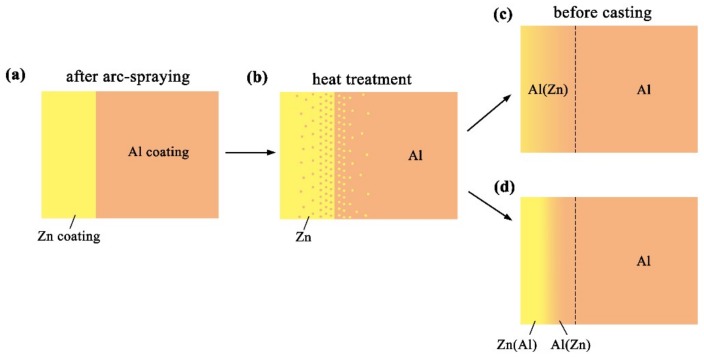
Schematic illustrations of the Al/Zn double-deck coating preheat treatment.

**Figure 12 materials-12-03273-f012:**
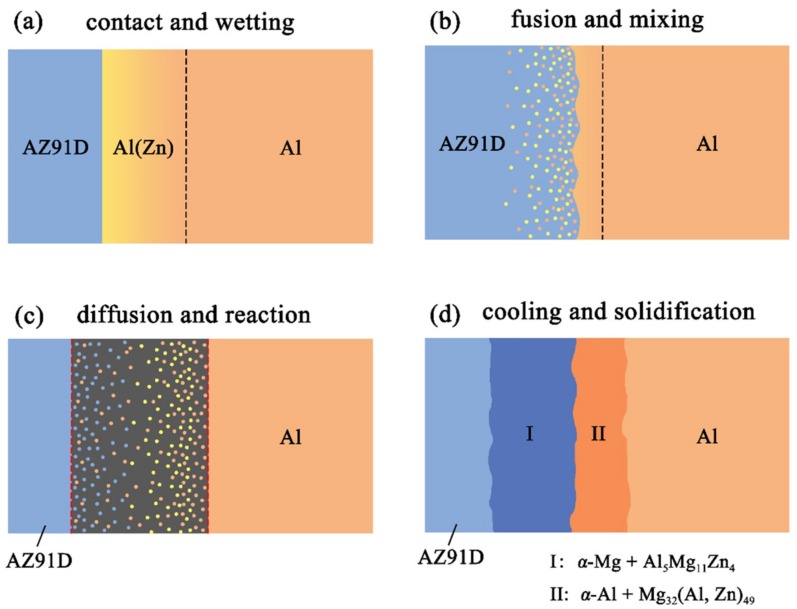
Schematic illustrations of the interface formation process in arc-sprayed Al/AZ91D with a Zn interlayer: The whole Al/Zn double-deck coating transformed into xAl-(1 − x)Zn coating (x > 50%).

**Figure 13 materials-12-03273-f013:**
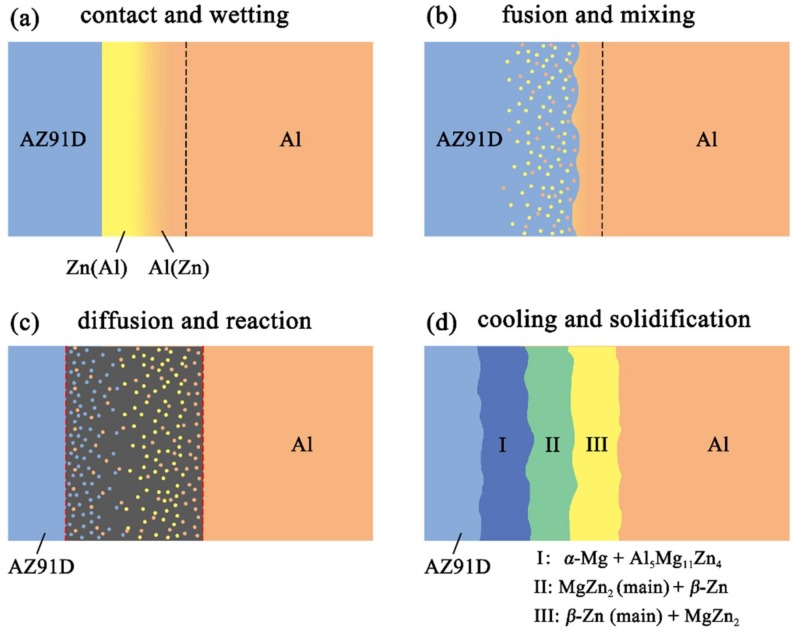
Schematic illustrations of the interface formation process in arc-sprayed Al/AZ91D with a Zn interlayer: Partial Al/Zn double-deck coating transformed into xAl-(1 − x)Zn coating (x > 50%).

**Figure 14 materials-12-03273-f014:**
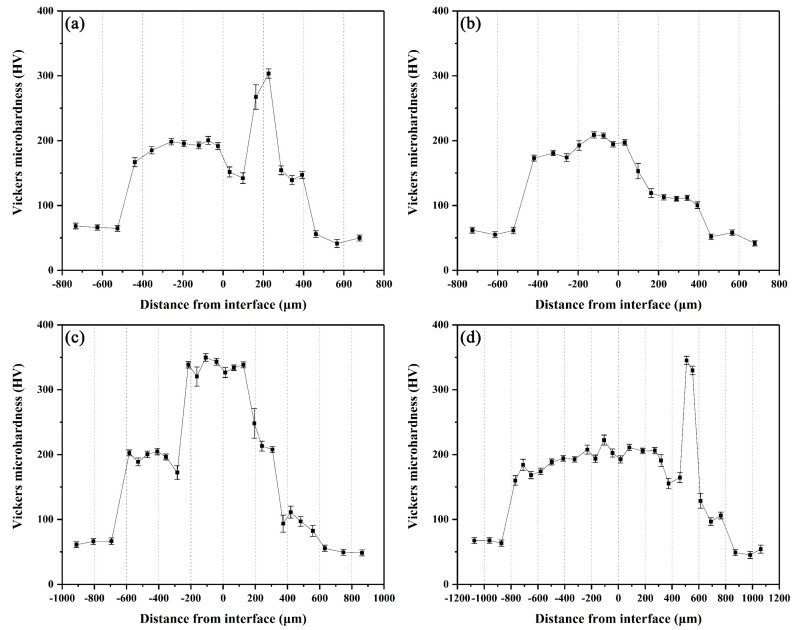
Vickers microhardness profiles across the interfaces of arc-sprayed Al/AZ91D with a Zn interlayer: (**a**) S18H6, (**b**) S18H12, (**c**) S30H6, and (**d**) S30H12.

**Figure 15 materials-12-03273-f015:**
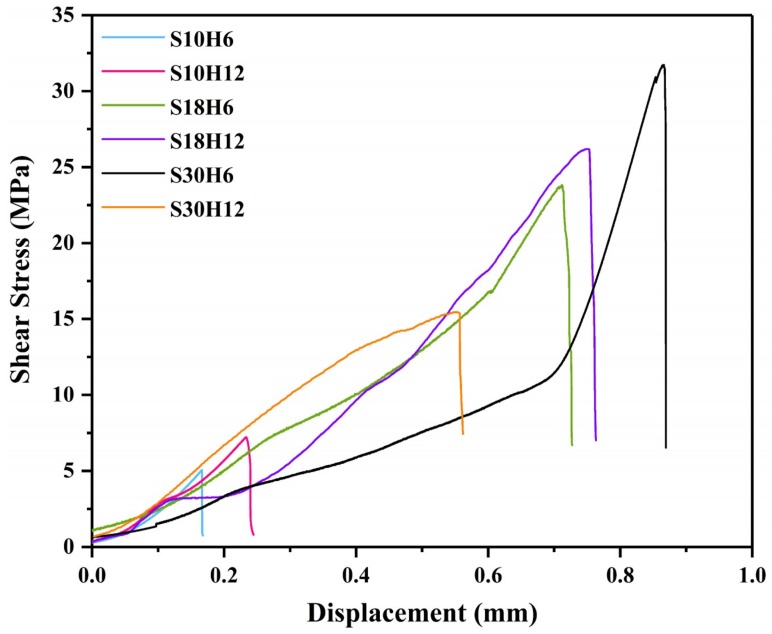
The relationship curves between the shear stress and deformation displacement of the arc-sprayed Al/AZ91D bimetals with a Zn interlayer from the shear test.

**Figure 16 materials-12-03273-f016:**
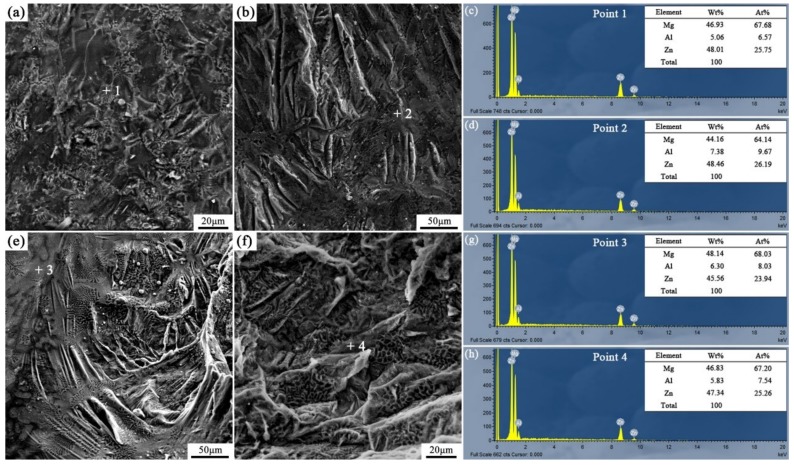
Fracture analysis of the arc-sprayed Al/AZ91D bimetals with a Zn interlayer: (**a**,**b**) SEM micrographs of S18H6 fracture surfaces, (**c**,**d**) EDS point scan spectra of points “1” and “2”, (**e**,**f**) SEM micrographs of S18H12 fracture surfaces, (**g**,**h**) EDS point scan spectra of points “3” and “4”.

**Figure 17 materials-12-03273-f017:**
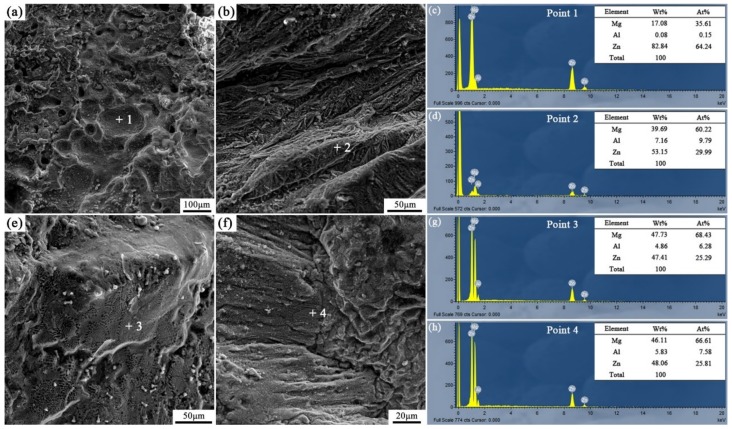
Fracture analysis of the arc-sprayed Al/AZ91D bimetals with a Zn interlayer: (**a**,**b**) SEM micrographs of S30H6 fracture surfaces, (**c**,**d**) EDS point scan spectra of points “1” and “2” (**e**,**f**) SEM micrographs of S30H12 fracture surfaces, (**g**,**h**) EDS point scan spectra of points “3” and “4”.

**Table 1 materials-12-03273-t001:** Chemical compositions of Al wire, Zn wire, and AZ91D ingot (wt.%).

Materials	Al	Zn	Mg	Fe	Cu	Mn	Si	Ni	Sn	Pb
Al wire	Bal	≤0.003	–	≤0.002	≤0.001	≤0.001	≤0.002	–	–	–
Zn wire	–	Bal	–	≤0.003	≤0.002	–	–	–	≤0.001	≤0.005
AZ91D ingot	9.580	0.6530	Bal	0.0017	0.0021	0.2194	0.0203	0.0001	–	–

**Table 2 materials-12-03273-t002:** Technological parameters of arc-spraying.

Technological Parameters	Value of Parameters	
	Aluminum Coating	Zinc Coating
Arc voltage (V)	34	27
Arc current (A)	150	70
Atomizing gas pressure (MPa)	0.6 ± 0.1	0.5 ± 0.1
Spraying distance (mm)	150 ± 20	100 ± 20
Feed voltage (V)	15	15

**Table 3 materials-12-03273-t003:** Process parameters of the arc-spraying of the Zn coating and preheat treatment of the Al/Zn double-deck coating.

Sample	Arc Spraying of Zn Coating		Preheat Treatment of Al/Zn Double-deck coating		Abbreviations
	Arc-spraying Zn Time (s)	Thickness of Zn Coatings (μm)	Preheat Temperature (K)	Preheat Time (h)	
1	10	250 ± 30	523	6	S10H6
2	10	250 ± 30	523	12	S10H12
3	18	440 ± 30	523	6	S18H6
4	18	440 ± 30	523	12	S18H12
5	30	720 ± 30	523	6	S30H6
6	30	720 ± 30	523	12	S30H12

**Table 4 materials-12-03273-t004:** EDS analysis results corresponding to the points marked in Figure 5.

Point No.	Element Compositions (at.%)				Possible Component	Point No.	Element Compositions (at.%)				Possible Component
	Mg	Zn	Al	O			Mg	Zn	Al	O	
1	53.67	31.86	14.47	–	Al_5_Mg_11_Zn_4_	8	36.49	58.69	4.82	–	MgZn_2_
2	92.52	1.28	6.20	–	*α*-Mg	9	36.07	58.50	5.43	–	MgZn_2_
3	66.94	26.86	6.20	–	*α*-Mg + Al_5_Mg_11_Zn_4_	10	4.96	87.50	7.54	–	*β*-Zn
4	80.83	15.09	4.08	–	*α*-Mg	11	1.17	47.34	46.01	5.48	*β*-Zn
5	60.70	30.91	8.39	–	*α*-Mg + Al_5_Mg_11_Zn_4_	12	5.18	26.32	65.66	2.84	*α*-Al + Mg_32_(Al, Zn)_49_
6	38.49	33.96	21.92	5.63	Al_5_Mg_11_Zn_4_	13	0.95	9.38	87.12	2.55	*α*-Al
7	44.29	42.41	13.30	–	Al_5_Mg_11_Zn_4_	14	0.45	4.99	88.91	5.65	*α*-Al

**Table 5 materials-12-03273-t005:** EDS analysis results corresponding to the points marked in Figure 6.

Point No.	Element Compositions (at.%)				Possible Component	Point No.	Element Compositions (at.%)				Possible Component
	Mg	Zn	Al	O			Mg	Zn	Al	O	
1	53.06	32.38	14.56	–	Al_5_Mg_11_Zn_4_	10	43.03	33.69	16.76	6.52	Al_5_Mg_11_Zn_4_
2	93.15	1.24	5.61	–	α-Mg	11	43.88	34.95	18.83	2.34	Al_5_Mg_11_Zn_4_
3	50.70	33.62	15.68	–	Al_5_Mg_11_Zn_4_	12	20.15	23.37	53.24	3.24	α-Al + Mg_32_(Al, Zn)_49_
4	78.53	15.51	5.96	–	α-Mg	13	22.44	25.26	46.78	5.52	α-Al + Mg_32_(Al, Zn)_49_
5	68.60	24.81	6.59	–	α-Mg + Al_5_Mg_11_Zn_4_	14	7.47	12.52	75.58	4.43	α-Al
6	65.81	25.42	8.77	–	α-Mg + Al_5_Mg_11_Zn_4_	15	1.85	6.40	89.67	2.08	α-Al
7	44.84	37.00	16.41	1.75	Al_5_Mg_11_Zn_4_	16	20.21	25.27	50.99	3.53	α-Al + Mg_32_(Al, Zn)_49_
8	65.63	22.69	5.90	5.78	α-Mg + Al_5_Mg_11_Zn_4_	17	0.74	5.22	89.41	4.63	α-Al
9	64.18	26.59	9.23	–	α-Mg + Al_5_Mg_11_Zn_4_						

**Table 6 materials-12-03273-t006:** EDS analysis results corresponding to the points marked in Figure 7.

Point No.	Element Compositions (at.%)				Possible Component	Point No.	Element Compositions (at.%)				Possible Component
	Mg	Zn	Al	O			Mg	Zn	Al	O	
1	58.97	17.56	23.47	–	Al_5_Mg_11_Zn_4_	11	33.40	56.62	2.39	7.59	MgZn_2_
2	89.39	1.16	4.33	5.12	*α*-Mg	12	35.55	53.73	6.35	4.37	MgZn_2_
3	48.40	35.27	16.33	–	Al_5_Mg_11_Zn_4_	13	36.92	58.39	4.70	–	MgZn_2_
4	79.10	16.46	4.44	–	*α*-Mg	14	63.36	30.56	6.09	–	*α*-Mg + Al_5_Mg_11_Zn_4_
5	69.25	25.97	4.78	–	*α*-Mg + Al_5_Mg_11_Zn_4_	15	48.77	43.62	7.61	–	Al_5_Mg_11_Zn_4_
6	49.81	33.60	11.32	5.27	Al_5_Mg_11_Zn_4_	16	0.75	89.85	2.10	7.29	*β*-Zn
7	36.76	53.70	8.23	1.31	MgZn_2_	17	32.86	59.61	1.73	5.80	MgZn_2_
8	33.48	57.68	3.08	5.76	MgZn_2_	18	4.10	94.30	1.60	–	*β*-Zn
9	5.05	85.53	2.05	7.37	*β*-Zn	19	36.95	60.43	2.62	–	MgZn_2_
10	4.89	93.17	1.95	-	*β*-Zn	20	0.29	6.64	93.07	–	*α*-Al

**Table 7 materials-12-03273-t007:** EDS analysis results corresponding to the points marked in Figure 8.

Point No.	Element Compositions (at.%)	Possible Component	Point No.	Element Compositions (at.%)	Possible Component
Mg	Zn	Al	O	Mg	Zn	Al	O
1	54.14	31.37	14.49	–	Al_5_Mg_11_Zn_4_	8	36.23	58.45	5.32	–	MgZn_2_
2	93.26	1.17	5.57	–	*α*-Mg	9	35.84	58.27	5.89	–	MgZn_2_
3	79.35	14.63	6.02	–	*α*-Mg	10	5.49	91.54	2.97	–	*β*-Zn
4	67.07	26.25	6.68	–	*α*-Mg + Al_5_Mg_11_Zn_4_	11	0.37	91.68	3.17	4.78	*β*-Zn
5	64.86	26.61	8.53	–	*α*-Mg + Al_5_Mg_11_Zn_4_	12	33.57	61.84	1.95	2.64	MgZn_2_
6	45.11	36.43	17.62	0.84	Al_5_Mg_11_Zn_4_	13	1.38	6.59	88.32	3.71	*α*-Al
7	61.30	30.76	7.94	–	*α*-Mg + Al_5_Mg_11_Zn_4_

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
