# Peer review of "Influence of a Zn Interlayer on the Interfacial Microstructures and Mechanical Properties of Arc-Sprayed Al/AZ91D Bimetals Manufactured by the Solid–Liquid Compound Casting Process"

_materials, 2019, doi:10.3390/ma12193273_

Round 1

Reviewer 1 Report

The paper deals with Influence of Zn interlayer on interfacial microstructures and mechanical properties of arc sprayed-Al/AZ91D bimetals manufactured by solid liquid compound casting process

The structure of the scientific report is good and well-understood. The aim is clarified. The introduction summarizes well the fabrication of bulk biomimetic materials with organized structures.

Discussions: In this study, it was a new exploration that the arc-spraying technique was used to prepare both solid matrix and metallic interlayer in the whole process. The arc-spraying time of Zn coating and the preheat time of Al/Zn double-deck coating were two main variables, which was in connection with the formation of microstructure and the evolution of the interface zone. In this current work was to investigate the effects of Zn interlayer on the interfacial microstructures and mechanical properties in arc-sprayed-Al/AZ91D bimetals by SLCC process.

Bibliographic references are appropriate. The author mentioned 21 literatures according to the reference list. These references are from the last 10 years, from prestigious journals.

The reviewer suggests to accept the paper for publication.

Author Response

Response to Reviewer 1 Comments

The structure of the scientific report is good and well-understood. The aim is clarified. The introduction summarizes well the fabrication of bulk biomimetic materials with organized structures.

Discussions: In this study, it was a new exploration that the arc-spraying technique was used to prepare both solid matrix and metallic interlayer in the whole process. The arc-spraying time of Zn coating and the preheat time of Al/Zn double-deck coating were two main variables, which was in connection with the formation of microstructure and the evolution of the interface zone. In this current work was to investigate the effects of Zn interlayer on the interfacial microstructures and mechanical properties in arc-sprayed-Al/AZ91D bimetals by SLCC process.

Bibliographic references are appropriate. The author mentioned 21 literatures according to the reference list. These references are from the last 10 years, from prestigious journals.

The reviewer suggests to accept the paper for publication.

Response: We really appreciate this reviewer for reviewing our manuscript and giving such high praise. During the revision stage, we will take this opportunity to fix some defects in the original manuscript and continue refining our work.

Reviewer 2 Report

This work is a thorough investigation of the process of making an Al coating on a Mg-based alloy with a Zn interlayer. The authors proposed using arc spraying in combination with a solid-liquid compound casting process. The study includes detailed structure characterization of the obtained layered systems by SEM/EDS and XRD.

Here are some suggestions to revise the paper before publication.

The Abstract should state the goal of the research and some general ideas underlying this work. Only after that a discussion of the particular values of the experimental parameters is appropriate.

Abbreviation "SLCC" should be explained not only in the Abstract but in the text as well.

Please provide more references to the basics of traditional arc spraying and SLCC.

Please revise the sentence "However, the maximum temperature in Al-Mg welding process is much higher than that in magnesium casting, and then it was difficult to realize the joint of solid metal to interlayer and joint of liquid metal to interlayer at the same time by SLCC" as it is rather hard to understand in the present form.

This sentence "Moreover, there are lots of metallurgical reactions and chemical reactions among Al, Mg and Zn elements under some specific conditions" should be revised ("lots of" is not a suitable phrase).

Line 88: "Table 1. In order to spray.." - ?

Fig. 1 is very good and helpful to understand the sequence of operations, could you add a text description of the whole process into the Abstract as well to make it easier for the reader to grasp the technological process of your work?

For hardness values, confidence intervals or standard deviation should be given.

Two decimal digits for the shear strength values - does this make sense? It would be reasonable to round up.

"Shouldn’t" - should be "should not".

Author Response

Response to Reviewer 2 Comments

Point 1: The Abstract should state the goal of the research and some general ideas underlying this work. Only after that a discussion of the particular values of the experimental parameters is appropriate.

Response 1: We appreciate this reviewer for providing the good advice. The sentence “The effect of Zn interlayer on microstructures, properties, and fracture behaviors of arc-sprayed-Al/AZ91D bimetals by SLCC was investigated and discussed in this study.” had been added in the updated Abstract part, and it stated and stressed the goal and some general ideas of our research.

Point 2: Abbreviation "SLCC" should be explained not only in the Abstract but in the text as well.

Response 2: We deeply thank this reviewer for the precious suggestion. The explanation for the abbreviation “SLCC” had been already added in lines 48 and 49 where the abbreviation “SLCC” first appeared in the text. Please see the details in the updated manuscript.

Point 3: Please provide more references to the basics of traditional arc spraying and SLCC.

Response 3: Thank this reviewer for giving the precious opinion. More references about the traditional arc spraying process and SLCC process had already been added into the introduction part of the updated manuscript.

Point 4: Please revise the sentence "However, the maximum temperature in Al-Mg welding process is much higher than that in magnesium casting, and then it was difficult to realize the joint of solid metal to interlayer and joint of liquid metal to interlayer at the same time by SLCC" as it is rather hard to understand in the present form.

Response 4: We are very grateful to this reviewer and quite agree on this nice suggestion. The sentence had been modified and rewritten as “The base and interlayer metals could be easily melted and mixed together under the effect of high temperature during the welding process. However, the Mg casting temperature is much lower than the Al-Mg welding temperature, and then it was difficult to realize the joint of solid metal to interlayer and the joint of liquid metal to interlayer at the same time by SLCC.”

Point 5: This sentence "Moreover, there are lots of metallurgical reactions and chemical reactions among Al, Mg and Zn elements under some specific conditions" should be revised ("lots of" is not a suitable phrase).

Response 5: Thank this reviewer very much for providing the good advice. In deed, “lots of” was not an appropriate phrase for the sentence here. As a result, “many kinds of ” had been used here instead of “lots of” in the revised manuscript.

Point 6: Line 88: "Table 1. In order to spray.." - ?

Response 6: We feel so sorry for the mistakes making this reviewer confused, and it was a layout problem. “Table 1.” here belonged to the previous sentence, and “In order to spray..” here belonged to the next sentence. It should be like this: “The chemical compositions of aluminum wires, zinc wires and AZ91D ingots are displayed in Table 1. In order to spray coatings on the cavity surfaces…”. We had made a modification in the updated manuscript.

Point 7: Fig. 1 is very good and helpful to understand the sequence of operations, could you add a text description of the whole process into the Abstract as well to make it easier for the reader to grasp the technological process of your work?

Response 7: We thank this reviewer for the valuable comment and suggestion. The sentence “The Al/Mg bimetal was produced by pouring the AZ91D melt into the moulds sprayed with Al/Zn double-deck coating, during which the arc-sprayed Zn coating acted as the interlayer.” had been added into the updated Abstract part, and it gave a brief introduction about the whole technological process.

Point 8: For hardness values, confidence intervals or standard deviation should be given.

Response 8: We are very grateful to this reviewer and quite agree on this nice suggestion. The error bars had been added into the updated Fig. 14, which could show the standard deviation of the microhardness values across the interface with the arc-sprayed-Al/AZ91D bimetals with Zn interlayer.

Point 9: Two decimal digits for the shear strength values - does this make sense? It would be reasonable to round up.

Response 9: It is a good idea to round up the shear strength values. However, we calculated the shear strength of the arc-sprayed-Al/AZ91D bimetals by using the two decimal digits of the maximum shear stress and contacted area. What’s more, it would be more convenient to have a direct comparison with the results of our previous study if the results could have a same decimal standard with the previous results. Thus, we decided to use two decimal digits for the shear strength values after a careful consideration. We thank this reviewer for the valuable suggestion all the same.

Point 10: "Shouldn’t" - should be "should not".

Response 10: Thank this reviewer very much for providing the good advice. All the “Shouldn’t” had been modified to “should not” in the updated manuscript. We had also checked the similar structures (for example, “couldn’t”, “weren’t” and “mustn’t”) and revised all the mistakes.

Reviewer 3 Report

The authors investigated the effects of Zn interlayer on interfacial microstructure and mechanical properties 505 of arc-sprayed-Al/AZ91D bimetals by SLCC.

Fig 14 (a), and (d) : Explain the reason why the sharp peak around 200 μm in (a), and 500-600 μm in (d). Fig. 15 : The lines for each samples are hard to distinguish with each other. Change the thickness of the line every samples, and, make it plain. L433-L439 : "...shear strength sharply dropped from 31.73 MPa to 15.44 437 MPa." ; Is there any cases or examples in this kind of materials?

Author Response

Response to Reviewer 3 Comments

Point 1: Fig 14 (a), and (d) : Explain the reason why the sharp peak around 200 μm in (a), and 500-600 μm in (d).

Response 1: As mentioned in the revised paper, the microhardness of MgZn2 intermetallic compound was about 327HV, which was higher than that of the other phases or compounds within the interface zone. The MgZn2 intermetallic compounds formed within the interface zones of S18H6, S18H12 and S30H6 samples. Although the amounts of MgZn2 intermetallic compounds in S18H6 and S18H12 samples were less than that in S30H6 sample, the layer of MgZn2 intermetallic compound still caused the sharp peaks around 200 μm in Fig 14 (a), and 500-600 μm in Fig 14 (d).

Point 2: Fig. 15 : The lines for each samples are hard to distinguish with each other. Change the thickness of the line every samples, and, make it plain.

Response 2: We thank this reviewer for giving the nice suggestion. The thicknesses and colors of the lines in Fig. 15 had been changed, and it was easier to distinguish the line for each sample in the updated figure.

Point 3: L433-L439 : "...shear strength sharply dropped from 31.73 MPa to 15.44 437 MPa." ; Is there any cases or examples in this kind of materials?

Response 3: There were two main reasons leading to the sharp drop of the shear strength from 31.73 MPa to 15.44 MPa, which had been discussed in this study. First, it was the increase of the intermetallic compounds with high brittleness, and the S30H12 sample had a bigger interface zone than the S30H6 sample. According to the studies about Al/Mg bimetallic materials by Li et al. [1] and Hajjari et al. [2], the shear strength would reduce with the increase of thickness of the interface zone. The thickness growth of the interface zone would increase the number of intermetallic compounds with high brittleness, which further increased the occurrence probability of brittle fracture, would cause the reduction of shear strength between AZ91D matrix and Al coating. Secondly, the dispersive distribution of Zn solid solution in MgZn2 intermetallic compounds also improved the shear strength of S30H6 sample, which reached an agreement with the study of Liu et al. [3]. The comprehensive effect of these two factors made that the shear strength of the S30H6 sample was much higher than that of the S30H12 sample.

References

[1] G. Li, W. Jiang, Z. Fan, Z. Jiang, X. Liu, F. Liu, Effects of pouring temperature on microstructure, mechanical properties, and fracture behavior of Al/Mg bimetallic composites produced by lost foam casting process, The International Journal of Advanced Manufacturing Technology 91(1) (2017) 1355-1368.

[2] E. Hajjari, M. Divandari, S.H. Razavi, S.M. Emami, T. Homma, S. Kamado, Dissimilar joining of Al/Mg light metals by compound casting process, Journal of Materials Science 46(20) (2011) 6491-6499.

[3] L.M. Liu, L.M. Zhao, Z.H. Wu, Influence of holding time on microstructure and shear strength of Mg–Al alloys joint diffusion bonded with Zn–5Al interlayer, Materials Science and Technology 27(9) (2011) 1372-1376.